Morphological identification and molecular confirmation of the deep-sea blue and red shrimp Aristeus antennatus larvae

Carreton Marta 1 mcarreton@icm.csic.es
Company Joan B. 1
Planella Laia 2
Heras Sandra 2
García-Marín José-Luis 2
Agulló Melania 2
http://orcid.org/0000-0002-2491-6203 Clavel-Henry Morane 1
Rotllant Guiomar 1
http://orcid.org/0000-0002-2238-9315 dos Santos Antonina 3
Roldán María Inés 2
1 Institut de Ciències del Mar (ICM-CSIC), Consejo Superior de Investigaciones Científicas , Barcelona , Spain
2 Laboratori d’Ictiologia Genètica, Universitat de Girona , Girona , Spain
3 Instituto Português do Mar e da Atmosfera (IPMA) , Lisboa , Portugal
Reimer James
Electronic publication date: 2019 Feb 12
Publication date: 2019
Volume: 7
Electronic Location ID: e6063
Received 2018 Aug 7; Accepted 2018 Nov 3
Copyright: © 2019 Carreton et al.
Copyright year: 2019
Copyright holder: Carreton et al.
License: This is an open access article distributed under the terms of the Creative Commons Attribution License, which permits unrestricted use, distribution, reproduction and adaptation in any medium and for any purpose provided that it is properly attributed. For attribution, the original author(s), title, publication source (PeerJ) and either DOI or URL of the article must be cited.
License URL: https://creativecommons.org/licenses/by/4.0/

Keywords: Aristeus antennatus, Molecular identification, SEM, Blue and red shrimp, Larvae

Funding: Spanish Ministerio de Economía y Competitividad (MINECO) Marta Carreton benefited from pre-doctoral FPU fellowship FPU2015 Spanish Ministerio de Educación, Cultura y Deporte Laia Planella benefited from predoctoral fellowship from the Universitat de Girona BR2014, Spain This research was carried out within project CONECTA (CTM2014-54648-C2) funded by the Spanish Ministerio de Economía y Competitividad (MINECO). Marta Carreton benefited from pre-doctoral FPU fellowship (FPU2015) from the Spanish Ministerio de Educación, Cultura y Deporte. Laia Planella benefited from predoctoral fellowship from the Universitat de Girona (BR2014), Spain. The funders had no role in study design, data collection and analysis, decision to publish, or preparation of the manuscript.

==============================
The early life stages of the blue and red shrimp Aristeus antennatus (Decapoda: Dendrobranchiata: Penaeoidea: Aristeidae) were described by Heldt in 1955 based on plankton samples, larval rearing and assumptions of species habitat. Even with adequate keys, identification of its first larval stages remained a difficult task due to the lack of specific morphological characters which would differentiate them from other Penaeoidea species. Larvae of Aristeus antennatus were collected in the continental slope off the Spanish Mediterranean coast in August 2016 with a neuston net and preserved in ethanol 96%. DNA from the larvae was extracted and the molecular markers Cytochrome Oxidase I and 16S rDNA were sequenced and compared to that of adults with the objective of confirming the previous morphological description. Then, we present additional information to the morphological description of Aristeus antennatus larval stages through scanning electron microscopy and molecular analysis. This represents the first documented occurrence of Aristeus antennatus larvae off the Catalan coast and sets the grounds for further work on larval ecology and population connectivity of the species, which is an important contribution to a more sustainable fishery.

Introduction

The deep-sea blue and red shrimp Aristeus antennatus (Risso, 1816; Decapoda: Dendrobranchiata: Penaeoidea: Aristeidae) is one of the most valuable fishing resources in the Mediterranean Sea. It is the main target of bottom trawlers along the coasts of Northwestern Africa, Portugal, Spain, France, Italy and Malta (www.fao.org). In the Spanish Mediterranean coast, it can represent up to 50% of the economical benefits for the fishermen associations (Maynou, 2008, DGPAM, 2017). Mature females aggregate at the continental shelf break in the summer (Sardà, Cartes & Norbis, 1994, Sardà, Maynou & Talló, 1997). Despite its social and economic relevance in the area, the knowledge about the species’ life cycle remains incomplete as its larval stages are still scarcely known. As this fishery progresses toward more integrative methods of stock assessment and management, filling the knowledge gap about the early life stages of Aristeus antennatus is a crucial step in the study of its dispersal and population connectivity.

In dendrobranchiate shrimps, the first larval stage hatching from the egg is usually a free-living nauplius. This stage has up to six substages depending on the species. It is followed by a variable number of zoeal stages often referred to as protozoea (early zoeae, with natatory antennules and antennae) and mysis (late zoeae, where the natatory function is assured by the pereiopods). The last mysis then metamorphoses into the first decapodid, which after a number of moults will come to settle in the adult habitat (Anger, 2001).

The description of the larval stages of Aristeus antennatus (Heldt, 1955) was based on 35 individuals caught in a plankton survey around the Balearic archipelago. The assumptions leading to the attribution of the larvae to Aristeus antennatus were based on the author’s extensive knowledge of Penaeoid larvae in the Mediterranean Sea. This morphological identification has never been confirmed since. To present date, occurrence of Aristeus antennatus larvae has only been detected in low numbers in plankton surveys off the Algerian and Portuguese Atlantic coasts, the Canary Islands and in the Balearic Sea, identified using Heldt’s larval descriptions (Seridji, 1971; Dos Santos, 1998; Carbonell et al., 2010; Landeira, 2010; Torres et al., 2013). The morphological description of a second mysis stage (Torres et al., 2013), also based on larvae caught in plankton, was the most recent addition to the larval series, with the last larval stages still remaining unknown. Rearing of larvae in the laboratory from berried females is a usual technique to accomplish the description of a complete larval cycle (Di Muzio, Basile & Pessani, 2018). This is not possible in the case of Aristeus antennatus due to the particularity of dendrobranchiate shrimps releasing their eggs directly to the water column, as opposed to being carried by females. To our knowledge, only one deep-sea Penaeoid larva from a plankton survey has been molecularly identified (Bracken-Grissom et al., 2012). The available descriptions of Penaeoid larvae from laboratory studies correspond only to species inhabiting shallow waters—as for instance the caramote prawn Penaeus kerathurus (Torkmen, 2003)—since females are easily cultured in tanks and eggs can be collected from the water upon release. No study has yet reported the culture of females of any deep-sea Penaeoid species.

The morphological identification of Penaeoidea first larval stage, the protozoea I (PZI), is often a particularly difficult task due not only to the small size of the specimens but also to the fact that in some cases different species share the same larval morphology (Martin, Criales & Dos Santos, 2014). It is only in later stages that the different larval series can be more easily distinguished based on morphological traits visible at the optical stereomicroscope, such as the presence and number of rostrum spines, supraorbital and/or pterygostomian spines, etc. On this matter, the early larval stages of Aristeus antennatus are no exception. In fact, the description of the PZI was based on a single individual captured in the plankton of Balearic waters, in the Western Mediterranean Sea (Heldt, 1955). The morphological characters that distinguish Aristeus antennatus PZI from other Penaeoid species present in our study area such as Sicyonia carinata, Parapenaeus longirostris or Funchalia woodwardii are generally clear (Dos Santos & Lindley, 2001). However, the distinction between Aristeidae and Benthesicymidae PZI stage is more difficult because the only morphological character that allows their differentiation is a small endopod on the third maxilliped (mxp3), present in Aristeus antennatus and absent in Gennadas spp. (Heldt, 1955; Gurney, 1924). In the case of Aristeus antennatus, the mxp3 is birramous, with two long plumose setae and one small simple seta on the exopod. In the case of Gennadas spp., the mxp3 is unirramous with two long plumose terminal setae. The reproductive period of the genus Gennadas has not yet been studied, but larvae caught in the plankton all year round have been classified as Gennadas spp. according to available information (Fusté, 1982, 1987; Torres et al., 2014). The reproductive period of Aristeus antennatus in the Mediterranean Sea is strictly seasonal in the summer (Company et al., 2003). With both larval types occurring concurrently during the summer, their morphological differentiation becomes a key issue in the study of decapod larval communities and of Aristeus antennatus larval distribution in particular.

The available descriptions of penaeoid larvae from plankton surveys have generally based their identification on an extensive knowledge of the adult morphology and ecology (Heldt, 1938, 1955; Gurney, 1924). Although morphological identification is an essential first step, in some cases it can be insufficient and lead to misidentifications (Palero, Guerao & Abelló, 2008; Sullivan & Neigel, 2017). In this context, the use of molecular markers can be particularly useful in the confirmation of visual identification of specimens, in complement to keys and descriptions based on morphological characters (Olson, Runstadler & Kocher, 1991; Webb et al., 2006). Previous studies on decapod crustacean larvae have used this technique either to confirm existing descriptions or as a complement to the descriptions of new stages (Raupach & Radulovici, 2015; Landeira et al., 2014; Bracken-Grissom et al., 2012; Pan et al., 2008).

The objective of this study was to examine the morphology of the first protozoea of the deep-sea blue and red shrimp Aristeus antennatus in order to find useful characters to distinguish it from Gennadas spp. larvae and to use molecular techniques to confirm the identification of all its known larval stages.

Materials and Methods

Sampling and morphological identification of Aristeus antennatus and Gennadas spp. larvae and adult specimens

In order to obtain both Aristeus antennatus and Gennadas spp. larval types, we performed two plankton samplings, one in the summer when Aristeus antennatus is at its peak reproductive period, and one in the winter, when Aristeus antennatus does not reproduce but Gennadas spp. larvae are likely to be found. Nevertheless, the presence of Gennadas spp. larvae has been reported in the summer in the Balearic Sea (Torres et al., 2014) and we were aware that we could encounter a mix of both species when aiming to collect Aristeus antennatus PZI larvae in the summer.

Summer sampling took place during a deep-sea cruise from mid-July to the end of August 2016 on board the research vessel García del Cid in various locations off the Spanish Mediterranean coast (Fig. 1). Plankton samples were taken using a 0.5 m2-mouth neuston net with a 300-μm mesh between 0.5 and one m depth over bottoms of 123–1,626 m. The samples were rinsed with distilled water and preserved in 96% ethanol. Samples were sorted in the laboratory using a Leica Wild M6 stereomicroscope and all larvae morphologically identified as Aristeus antennatus following the available descriptions (Heldt, 1955; Torres et al., 2013) were stored individually in 96% ethanol.

Figure 1 Stations where larvae were selected.

Red dots: summer sampling. Blue dots: winter sampling. Bathymetry is shown every 200 m.

Winter sampling took place from mid-February to early March 2017 during a deep-sea cruise on board the same research vessel off the NW Mediterranean coast (Fig. 1). Plankton samples were taken in integrated oblique tows using a 60-cm diameter bongo with a 300-μm mesh net between 500 m depth and the surface, over bottoms of 1,952 and 1,790 m. They were sorted on board using an Olympus SZ stereomicroscope and larvae morphologically identified as Gennadas spp. PZ I following the available description (Gurney, 1924) were rinsed with distilled water and preserved individually in 96% ethanol.

Identifications of larvae through DNA barcoding are only reliable when the obtained sequences are compared to those of adult specimens of known species. Adults of Aristeus antennatus from commercial trawling vessels had been previously identified and cross-checked with the available literature (Zariquiey-Álvarez, 1968). They had been collected, preserved and their DNA amplified and the resulting sequences were therefore available for comparison. The GenBank accession numbers are EU977139–EU977140 for 16S rDNA (Sardà et al., 2010) and EU908514 for Cytochrome Oxidase I (COI) (Roldán et al., 2009). In the case of Gennadas elegans, two adult individuals were selected from a sampling cruise in May 2010 in the Mediterranean Sea. They had been previously identified by Dr. Pere Abelló at the Institut de Ciències del Mar according to the available literature (Zariquiey-Álvarez, 1968) and preserved in 96% ethanol. For the purpose of this paper, we extracted DNA from their abdominal tissue and sequenced the product following the same method as for Gennadas spp. larvae to provide genetic information about the species (GenBank accession numbers MH605176 and MH605177).

Analysis of the protozoea I pool

Morphological analysis

In order to closely examine the morphology of the PZI larvae from the summer sampling, scanning electron microscopy (SEM) was used for 10 randomly selected individuals identified as Aristeus antennatus PZI. Also, three individuals from the winter sampling, morphologically identified as Gennadas spp. PZI, were randomly selected. Both sets of larvae were immersed in a graded acetone series (25%, 50%, 75% and 100%), dried to critical point, mounted on stubs with self-adhesive carbon stickers and coated in gold. They were observed under a Hitachi S-3500N SEM.

Furthermore, measurements of carapace length, telson rami length, telson angle and length of last somite of pleon were taken for all remaining individuals from the winter sampling identified as Gennadas spp. (n = 9). Also, the same measurements were carried out for 10 individuals from the summer sampling, identified as Aristeus antennatus. To do this, we used a Leica M205 C stereomicroscope and ImageJ image analysis software.

Molecular analysis

In order to confirm the identity of a representative sample of the PZI pool found in the summer sampling, we randomly selected 24 PZI individuals attributed to Aristeus antennatus, from a total of 527 found. Selection was done according to spatial criteria with the objective of covering the whole study area. A maximum of three larvae per station were selected where the total number found was highest. We also randomly selected four PZI larvae from the winter sampling, morphologically attributed to Gennadas spp., from a total of 11 individuals found. Information about larvae analyzed and their GenBank accession numbers are shown in Table 1.

Table 1 List of protozoea I larvae analyzed by station.

Stage	Station	Lon (°E)	Lat (°N)	Bottom depth (m)	Sampling depth (m)	N	Putative species	
PZI	56	0.7158	38.9593	858	0.5–1	1	A. antennatus	
60	0.4811	39.0395	800	0.5–1	1	A. antennatus	
98	1.3945	40.7777	404	0.5–1	2	A. antennatus	
105	1.8895	40.958	1,143	0.5–1	2	A. antennatus	
112	2.5133	41.298	695	0.5–1	1	A. antennatus	
113	2.5473	41.1702	1,038	0.5–1	1	A. antennatus	
115	2.5735	41.3575	300	0.5–1	1	A. antennatus	
120	3.0187	41.2557	1,473	0.5–1	3	A. antennatus	
138	3.5032	41.4412	1,380	0.5–1	1	A. antennatus	
126	3.1165	41.4957	331	0.5–1	1	A. antennatus	
123	2.9417	41.508	507	0.5–1	3	A. antennatus	
122	2.9698	41.4127	642	0.5–1	1	A. antennatus	
124-2	2.8203	41.5606	378	0.5–1	1	A. antennatus	
133-1	3.3415	41.8232	650	0.5–1	1	A. antennatus	
133-3	3.3538	41.9002	298	0.5–1	1	A. antennatus	
145	3.4609	42.2327	123	0.5–1	1	A. antennatus	
145-1	3.3855	42.3362	462	0.5–1	1	A. antennatus	
148	3.6878	41.8280	1,626	0.5–1	1	A. antennatus	
TOTAL						24		
PZI	A09	3.4130	41.2550	1,952	0–500	2	Gennadas spp.	
B05	2.8811	41.3418	1,790	0–500	2	Gennadas spp.	
TOTAL						4		
Note:

PZ, protozoea, N, number of individuals analyzed.

DNA isolations were carried out using a commercial kit optimized for small samples (Quick-DNA Microprep Plus kit, Zymo Research, Irvine, CA, USA) and resuspended in a final volume of 10 μL. A negative control that contained no sample was included in every isolation round to check for contamination during the experiments.

For larvae morphologically identified as Aristeus antennatus, a 617-base pair fragment of the mitochondrial gene COI was amplified by polymerase chain reaction (PCR) using the primer pair COILAa (5′ GGT GAC CCA GTC CTT TAC CA 3′) and COIHAa (5′ GTC TGG ATA ATC AGA ATA CCG AC 3′) (Roldán et al., 2009), specific for Aristeus antennatus. For larvae and adult individuals identified as Gennadas spp., a 658-base pair fragment of the COI gene was amplified using the primer pair CrustDF1 (5′ GGT CWA CAA AYC ATA AAG AYA TTG G 3′) (Steinke, Prosser & Hebert, 2016) and HCO-2198 (5′ TAA ACT TCA GGG TGA CCA AAA AAT CA 3′) (Folmer et al., 1994). PCRs were carried out in a final volume of 25 μL, containing 12.50 μL of Supreme NZYTaq Green PCR Master Mix (NZYTech, Lisboa, Portugal), 0.5 μM of each primer, 2.5 μL of the template DNA solution, and PCR-grade water up to 25 μL. The thermal cycling conditions were as follows: an initial denaturation step at 95 °C for 5 min, followed by 35 cycles of denaturation at 95 °C for 30 s; annealing at 53 °C (COILAa and COIHAa) or at 49 °C (CrustDF1 and HCO-2198) for 30 s; extension at 72 °C for 45 s; and a final extension step at 72 °C for 5 min. A negative control that contained no DNA was included in every PCR round to check for cross-contamination.

PCR products were run on a 1% agarose gel stained with Real Safe (Durviz, Paterna, Spain) and imaged under UV light, to verify amplicon size. PCR products were bidirectionally sequenced using the PCR primers. Electropherogram analysis and overlapping was conducted in Geneious 8.1.8 (Biomatters Ltd., Auckland, New Zealand). During electropherogram analysis, the primer annealing regions and the low quality regions at both ends of each electropherogram were trimmed (error probability limit of 0.03). Sequence reads were manually checked for sequencing errors or ambiguous base calls. The positions with double peaks were coded using the IUPAC ambiguity code (e.g., R: G or A). In order to check for possible pseudogenes, the sequences were aligned with BioEdit and no insertions or deletions were detected. Then, all sequences were translated into proteins with online software ExPASY (Gasteiger et al., 2003) and no stop codons were detected in the appropriate reading frame. The resulting nucleotide sequences were compared to available information in GenBank using Basic Local Alignment Search Tool (BLASTN 2.8.0, Zhang et al., 2000).

Analysis of all known larval stages of Aristeus antennatus

For the molecular confirmation of the rest of known larval stages of Aristeus antennatus, we randomly selected three PZII, one PZIII, three mysis I (MI) and one mysis II (MII) and followed the same procedure as in section Morphological analysis (Table 2).

Table 2 List of larvae analyzed for DNA regions COI (top) and 16S rDNA (bottom) by station.

Molecular marker	Stage	Station	Lon (°E)	Lat (°N)	Bottom depth (m)	Sampling depth (m)	N	
COI	PZI	(see Table 1)	
PZII	96	1.4862	40.6578	940	0.5–1	1	
138	3.5032	41.4412	1,380	0.5–1	1	
145	3.4609	42.2327	1,626	0.5–1	1	
PZIII	57	0.9092	38.9843	728	0.5–1	1	
MI	57	0.9092	38.9843	728	0.5–1	2	
96	1.4862	40.6578	940	0.5–1	1	
MII	57	0.9092	38.9843	728	0.5–1	1	
TOTAL		8	
16S rDNA	PZI	112	2.5133	41.298	695	0.5–1	3	
113	2.5473	41.1702	1,038	0.5–1	2	
PZII	124-1	2.8917	41.6367	200	0.5–1	1	
143b	3.4248	41.9788	187	0.5–1	2	
145	3.4609	42.2327	123	0.5–1	1	
133	3.2760	41.8762	600	0.5–1	2	
PZIII	57	0.9092	38.9843	728	0.5–1	2	
44	0.9008	38.6558	528	0.5–1	3	
MI	53	0.5502	38.9512	776	0.5–1	1	
57	0.5502	38.9843	728	0.5–1	1	
44	0.9008	38.6558	528	0.5–1	4	
MII	57	0.9092	38.9843	728	0.5–1	2	
TOTAL						24	
Notes:

Putative species is A. antennatus in all cases.

PZ, protozoea; M, mysis; N, number of individuals analyzed.

In addition, a molecular analysis with ribosomal gene 16S rDNA was conducted for all known larval stages. To do this, we randomly selected five PZI, six PZII, five PZIII, six MI and two MII (Table 2). Genomic DNA isolation from whole larvae (HotSHOT) was performed following Montero-Pau, Gómez & Muñoz (2008) with slight modifications. (PCR; Saiki et al., 1988) methods for amplification of the mitochondrial 16S rDNA gene followed the procedures outlined in Roldán et al. (2009). Standard precautions were adopted to detect contamination and related problems. PCR products were verified on 1% agarose gel with ethidium bromide (0.5 mg/mL) and were purified for sequencing by treating with exonuclease I and shrimp alkaline phosphatase (Werle et al., 1994). DNA sequencing reactions were carried out with BigDye Terminator v3.1 Cycle Sequencing Kit (Applied Biosystems, Foster City, CA, USA) according to the manufacturer’s instructions. Primers used for sequencing were the same as those employed for PCR amplifications. Finally, labelled fragments were loaded onto an ABI PRISM 3130 Genetic Analyzer (Applied Biosystems, Foster City, CA, USA) at the Laboratori d’Ictiologia Genètica, Universitat de Girona, Spain.

Nucleotide sequences were aligned and edited in Geneious v7.1.4 (Kearse et al., 2012). In order to confirm the identification of the larvae, simultaneous comparisons were done with reference sequences from adults of six species of dendrobranchiate decapod crustaceans: Aristeus antennatus, Aristaeomorpha foliacea, Gennadas elegans, Gennadas valens, Parapenaeus longirostris and Penaeus (Melicertus) kerathurus. The corresponding GenBank accession numbers are: EU977139–EU977140 (Sardà et al., 2010), MF496984–MF496986 (Roldán et al., 2017), JX403858.1 (Bracken-Grissom et al., 2012), EF589715.1 (Pascoal et al., 2008) and EU430762.1 (Zitari-Chatti et al., 2009), respectively.

Results

Analysis of the protozoea I pool

Morphological analyses

The morphology of the third maxilliped (mxp3) of the PZI larvae selected was studied under SEM. For the summer larvae examined (expected to be Aristeus antennatus), the mxp3 was not visible in two of the 10 selected individuals due to the specimens’ position. In the other eight, the mxp3 was clearly birramous and showed two long plumose setae on the exopod (Fig. 2). No small simple seta was observed on the exopod of the mxp3 in any of the individuals. For the winter larvae examined (then designed as Gennadas spp. since Aristeus antennatus only reproduces in summer), the mxp3 was clearly visible in two of the three individuals and it was birramous in both of them (Fig. 3). The endopod in Gennadas spp. is smaller and less conspicuous than that of Aristeus antennatus, but differences among individuals do not allow the use of this character for its taxonomical identification.

Figure 2 General view (A) and detail of a third maxilliped (B) of an Aristeus antennatus protozoea I.

Figure 3 General view (A) and detail of a birramous third maxilliped (B) of a Gennadas elegans protozea I.

Other morphological characters were observed under SEM, such as the presence of frontal organs and the telson angle, with no conclusive distinctive traits between both species. A comparison of all morphological characters observed is presented in Table 3.

Table 3 Comparison of relevant morphological characters of the first protozoea stage examined.

Morphological characters	Aristeus antennatus Heldt (1955)	Aristeus antennatus Present study	Gennadas sp. Gurney (1924)	Gennadas elegans Present study	
Total length (TL; mm)	1.55	1.20 ± 0.05	0.98	1.10 ± 0.10	
Carapace length (CL; mm)	NA	0.42 ± 0.03	NA	0.38 ± 0.04	
Mxp3	Birramous	Birramous	Unirramous	Birramous	
Number of setae exop. Mxp3	Two long plumose, one small simple	Two long plumose	Two long plumose	Two long plumose	
Frontal organs	Small, round	Small, round	Small, round	Small, round	
Length of the last somite of the pleon with telson (mm)	NA	0.38 ± 0.07	NA	0.32 ± 0.04	
Length of the telson rami (mm)	NA	0.11 ± 0.01	NA	0.11 ± 0.02	
Proportion: length of last somite of pleon with telson/length of telson rami	NA	0.30 ± 0.5	NA	0.35 ± 0.05	
Telson angle (°)	NA	35.73 ± 8.32	NA	35.84 ± 7.67	
Note:

Mxp3, third maxilliped.

Molecular analyses

The COI sequences from the 24 PZI larvae from the summer sampling revealed eight different haplotypes and all retrieved only Aristeus antennatus sequences when analyzed in BLAST (GenBank accession numbers MH605140–MH605163). To our knowledge, this is the first record of Aristeus antennatus PZI stage since its description from a single individual (Heldt, 1955). On the other hand, all four larvae morphologically identified as Gennadas spp. from the larval description showed an average resemblance of 99.38% to the sequences from the two G. elegans adult individuals analyzed (GenBank accession numbers MH605172–MH605175 for larvae; MH605176 and MH605177 for adults).

Molecular analysis of all known larval stages of Aristeus antennatus

All known stages of Aristeus antennatus were genetically identified with two markers: COI and 16S rDNA (Table 2). For marker COI, all eight larvae analyzed corresponding to the four remaining known larval stages of Aristeus antennatus were successfully sequenced (617 bp) and were identified as Aristeus antennatus. The sequences corresponded to two different haplotypes (GenBank accession numbers MH605164–MH605171). For marker 16S rDNA, all 24 larvae analyzed from the five known larval stages of the species were successfully sequenced (300 bp) and all were also undoubtedly identified as Aristeus antennatus. The sequences corresponded to four different haplotypes (GenBank accession numbers MH433629, MH433630, MH433631 and MH433632). Overall, we obtained a 309 bp global alignment with 84 variable positions that allow the discrimination of Aristeus antennatus from other Mediterranean Dendrobranchiata species.

Discussion

The larvae collected in the present study are the highest number of Aristeus antennatus larvae ever found in plankton samples, and analysis of these data (under preparation) could shed light on the species’ larval distribution and connectivity for the Western Mediterranean Sea. All collected larval stages of Aristeus antennatus were identified with the available morphological descriptions made by Heldt (1955) and Torres et al. (2013) and confirmed by molecular methods.

From a morphological point of view, there are still some difficulties in identifying the first prozoea of Aristeus antennatus, since this stage is strikingly similar to that of Gennadas spp. In this study, it was not possible to confirm the previous description of these two species, even with the use of SEM. The available description for the first prozoea of Gennadas spp. (Gurney, 1924) notes an unirramous mxp3, whereas in our case two of the three G. elegans PZI showed a small endopod on the mxp3. The presence of a birramous mxp3 in G. elegans PZI stage may eliminate the possibility of distinguishing it from Aristeus antennatus using morphological characters. The examination of other morphological traits such as telson invagination and anal spines structure did not yield any concluding results on features to differentiate both species despite the slight differences noted. On the other hand, the observed Aristeus antennatus PZI larvae invariably showed only two long plumose setae on the exopod of the mxp3. The description appointed, in addition, one small simple seta on the exopod of the mxp3 (Heldt, 1955), which was not observed in any of the individuals examined. This could suggest that the small simple seta described may appear later in the development of this stage and that newly molted PZI of Aristeus antennatus would only present the two long plumose setae. Nevertheless, we have examined all 527 Aristeus antennatus PZI individuals collected and all of them show only the two long plumose setae on the exopod of the mxp3 when observed at the stereomicroscope. The presence of a small simple seta on the exopod of the mxp3 is often used as a distinctive character of Aristeus antennatus PZI when the endopod is not clearly visible. However, in the light of our results we recommend that this feature not be used to distinguish the PZI of Aristeus antennatus and Gennadas elegans.

Our results show that 100% of the analyzed PZI larvae with the shared morphology of Aristeus antennatus/Gennadas spp. were molecularly identified as Aristeus antennatus. Nevertheless, the presence of Gennadas spp. older larval stages in this sampling collection (Carreton et al., 2018, unpublished data) calls for caution during morphological identification. Until a new and standardized description of both larval types is completed and a morphological character is found to tell both species apart, molecular techniques remain the most reliable method for a correct species identification. According to the descriptions of (Heldt, 1955), the larval series of Aristeus antennatus is also morphologically very similar to that of Aristaeomorpha foliacea (Risso, 1822). Both species keep a smooth abdomen and carapace throughout their larval cycle, and it is possible that their PZIs can be confused as well in areas where both species coexist abundantly (Central and Eastern Mediterranean). Unfortunately, there is still no available description of the PZI of Aristaeomorpha foliacea that we can use to clarify this matter. In our study area, records of Aristaeomorpha foliacea adult individuals have been decreasing and the species has been considered locally extinct in the Northwestern Mediterranean (Cau et al., 2002, Cartes, Maynou & Fanelli, 2011). Nevertheless, a more thorough examination of the available bibliography is being carried out to clarify the morphological differences among these larval forms in the hopes of facilitating a more accurate identification (Carreton et al., 2018, unpublished data).

Conclusions

Molecular analysis performed for all known larval stages of Aristeus antennatus caught in the Western Mediterranean Sea proved that available descriptions for Aristeus antennatus larval series are accurate.

Molecular tools used clearly differentiate PZI of Aristeus antennatus and Gennadas elegans.

From the morphological examination of the first protozoea stage, and contrary of what was expected, the mxp3 is similar for both Aristeus antennatus and Gennadas elegans, which makes this character not suitable for use in morphological identification.

The absence of a small simple seta on the exopod of the mxp3 of Aristeus antennatus first protozoeae examined indicates that this character is not suitable for use in the morphological differentiation of Aristeus antennatus and Gennadas elegans first protozoea.

Supplemental Information

Supplemental Information 1 Raw data: COI sequences (in FASTA format) for the Aristeus antennatus PZ I larvae analyzed in this study.

Click here for additional data file.

Supplemental Information 2 Raw data: COI sequences (in FASTA format) for Gennadas elegans larvae and adult individuals analyzed in this study.

Click here for additional data file.

Supplemental Information 3 Raw data: COI sequences (in FASTA format) for Aristeus antennatus larvae from stages PZII, PZIII, MI and MII analyzed in this study.

Click here for additional data file.

Supplemental Information 4 Raw data: 16S rDNA sequences (in FASTA format) for Aristeus antennatus larvae analyzed in this study.

Click here for additional data file.

The authors would like to thank Dr. Pere Abelló for providing and identifying the G. elegans adult individuals used in this study. Thanks to José Manuel Fortuño for his dedication and expertise during the SEM sessions. Molecular analyses of COI region were performed at AllGenetics (A Coruña, Spain).

Additional Information and Declarations

Competing Interests

Author Contributions

DNA Deposition

Data Availability

Antonina dos Santos is an Academic Editor for PeerJ. The rest of the authors declare that they have no competing interests.

Marta Carreton conceived and designed the experiments, performed the experiments, analyzed the data, prepared figures and/or tables, authored or reviewed drafts of the paper, approved the final draft.

Joan B. Company conceived and designed the experiments, authored or reviewed drafts of the paper, approved the final draft.

Laia Planella performed the experiments, analyzed the data, contributed reagents/materials/analysis tools, approved the final draft.

Sandra Heras conceived and designed the experiments, performed the experiments, analyzed the data, contributed reagents/materials/analysis tools, authored or reviewed drafts of the paper, approved the final draft.

José-Luis García-Marín conceived and designed the experiments, performed the experiments, analyzed the data, contributed reagents/materials/analysis tools, authored or reviewed drafts of the paper, approved the final draft.

Melania Agulló performed the experiments, analyzed the data, contributed reagents/materials/analysis tools, approved the final draft.

Morane Clavel-Henry conceived and designed the experiments, authored or reviewed drafts of the paper, approved the final draft.

Guiomar Rotllant conceived and designed the experiments, analyzed the data, authored or reviewed drafts of the paper, approved the final draft.

Antonina dos Santos analyzed the data, prepared figures and/or tables, authored or reviewed drafts of the paper, approved the final draft.

María Inés Roldán conceived and designed the experiments, performed the experiments, analyzed the data, contributed reagents/materials/analysis tools, authored or reviewed drafts of the paper, approved the final draft.

The following information was supplied regarding the deposition of DNA sequences:

The COI and 16S rDNA sequences obtained from this analysis are accessible via GenBank accession numbers MH605140–MH605177 (for COI), and MH433629–MH433632 (for 16S rDNA).

The following information was supplied regarding data availability:

The raw data are the DNA sequences and the SEM images in the article.

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
