# Peer review of "Morphological identification and molecular confirmation of the deep-sea blue and red shrimp Aristeus antennatus larvae"

_PeerJ, doi:10.7717/peerj.6063_

## Round 0.1 · original submission · Minor Revisions

I have heard back from two reviewers, who were generally positive about your work while recommending many constructive improvements. In particular, please pay attention to reviewer 1's third comment on experimental design; this must be addressed in any revision.

I look forward to receiving your revised version.

·

Basic reporting

The manuscript is very well written and fills an important knowledge gap in decapod morphology and ecology. The identification of larval stages across decapods is still lacking for most species, so these types of papers that use BOTH morphological and molecular methods for larvae-adult matching are very important. I would suggest the authors add the reference below to line 88-89.

Bracken-Grissom, H.D., Felder, D.L., Volmer, N., Martin, J. and K.A. Crandall. 2012. Phylogenetics Links Monster Larva to Deep-Sea Shrimp. Ecology and Evolution. 2(10): 2367-2373.

Experimental design

The experimental design was well-designed and executed. I would like to see a few additions before this paper is accepted for publications:
1. Why did the authors use different genes (COI vs. 16S) for the protozoea I pool and all known larval stages? Both are sufficient for DNA barcoding, however I find it strange to switch from one marker to the next. An explanation should be provided.
2. As I am sure the authors are familiar with the presence of pseudogenes, especially within COI, so I would insist there is mention to how the authors examined the sequences for the presence and removal of pseudogenes in the COI data.
3. Lastly, and most importantly, the larval-adult matching studies only work if a reliable ADULT specimen is available for larval matching. GenBank is absolutely rampant with identification errors. I would like to see the authors provide an expanded and clearly written explanation as to why they are 100% confident in the adult identifications used to match the larvae. I noticed that a co-author is cited at the end of line 195 (also on the present paper) but an expanded justification about the morphological confirmation of adults is needed. Also, several of the GenBank numbers provided in the line 194-195 are either 1) unpublished or 2) not released. This also makes me feel very uneasy, without expansion on this. For example, JX403858 is NOT from Roldan et al. so the appropriate papers needs to also be cited. I do warn caution from my own experience.

Validity of the findings

Once an expanded justification on the adults identification is expanded the findings are valid.

Additional comments

Overall, I think this is a very valuable paper that can advance the field of fisheries and decapod taxonomy.

Reviewer 2 ·

Basic reporting

I was interested in reading the manuscript PeerJ-29455 regarding the morphological and molecular confirmarion of the larval stage PZI of Aristeus antennatus. The topic is interesting and the approach, based on mprphological and molecular studies offer valuable offer to the literature.

The low number of the individuals studied may consist a controversy among the reviewers and other researchers, but - according me - this number reflects the general population's
situation.

Some references are too old and I suggest to authors to replace them with more recent ones.

Experimental design

The experimental design is efficient and according the more recent experimental design. The unique gap is the depth of each sampling station (please insert a coloumn in Table 1 indicating the depth of each station). Did the plankton samples of A. antennatus taken from 0,5-1 m?
An other point is the missing of Gennadas spp samples in summer. For comparative reasons the coexistence of both species caught in the same period could improve the results of the manuscript.

Validity of the findings

In general, it is an interesting manuscript providing novel, detailed information on the larval stages of both species (A. antennatus and G. elegans) and improves the knowledge to the taxinomists and marine biologists. The Discussion is well documented and well stractured.

Additional comments

The paper, however, is unclear in some lines and needs to be clarified before publication. For example, these lines are reported to the depth stratum of the sampling stations, on the absence of Gennadas spp samplings from the summer surveys, some changes in the Introduction.

Replace some references which are old

Annotated reviews are not available for download in order to protect the identity of reviewers who chose to remain anonymous.

---

## Round 0.2 · accepted · Accept

I have gone over the manuscript, and it has been well-revised. Thank you for your hard work, and I look forward to seeing the published version of this work.

#